# Fostering Trustworthiness of Federated Learning Ecosystem through Realistic Scenarios

**Athanasios Psaltis** [1,2,*], **Kassiani Zafeirouli** [1], **Peter Leškovský** [3], **Stavroula Bourou** [4], **Juan Camilo Vásquez-Correa** [3], **Aitor García-Pablos** [3], **Santiago Cerezo Sánchez** [3], **Anastasios Dimou** [1], **Charalampos Z. Patrikakis** [2] and **Petros Daras** [1]

[1] Centre for Research and Technology Hellas, 57001 Thessaloniki, Greece; cassie.zaf@iti.gr (K.Z.); dimou@iti.gr (A.D.); daras@iti.gr (P.D.)

[2] Department of Electrical and Electronics Engineering, University of West Attica, 12241 Athens, Greece; bpatr@uniwa.gr

[3] Vicomtech Foundation, Basque Research and Technology Alliance (BRTA), 20009 Donostia-San Sebastian, Spain; pleskovsky@vicomtech.org (P.L.); jcvasquez@vicomtech.org (J.C.V.-C.); agarciap@vicomtech.org (A.G.-P.); scerezo@vicomtech.org (S.C.S.)

[4] Synelixis Solutions S.A., 34100 Chalkida, Greece; bourou@synelixis.com

\* Correspondence: apsaltis@uniwa.gr; Tel.: +30-2310-464160

**Abstract:** The present study thoroughly evaluates the most common blocking challenges faced by the federated learning (FL) ecosystem and analyzes existing state-of-the-art solutions. A system adaptation pipeline is designed to enable the integration of different AI-based tools in the FL system, while FL training is conducted under realistic conditions using a distributed hardware infrastructure. The suggested pipeline and FL system's robustness are tested against challenges related to tool deployment, data heterogeneity, and privacy attacks for multiple tasks and data types. A representative set of AI-based tools and related datasets have been selected to cover several validation cases and distributed to each edge device to closely reflect real-world scenarios. The study presents significant outcomes of the experiments and analyzes the models' performance under different realistic FL conditions, while highlighting potential limitations and issues that occurred during the FL process.

**Keywords:** federated learning; trustworthiness; privacy-preserving technologies

## 1. Introduction

The effectiveness of artificial intelligence (AI) models is closely tied to the quality and quantity of data utilized in their training. With the abundance of information available on various devices, including mobile and servers, there is an opportunity to glean valuable insights. Federated learning (FL) is an innovative machine learning (ML) approach that leverages decentralized data and computational resources to deliver more tailored and flexible applications while upholding the privacy of users and organizations. By utilizing FL, it is possible to uphold data protection laws and regulations, thereby ensuring that privacy is not compromised. FL has demonstrated exceptional results in numerous analysis tasks, such as image classification, object detection, and action recognition [1–3], indicating its robustness and effectiveness in these areas. To elaborate further, FL allows data to remain on individual devices (cross-device) or servers (cross-silo) rather than being centralized, thereby avoiding potential privacy breaches. This approach enables users to keep their data safe and secure while still contributing to the training of ML models. Moreover, FL can handle large and diverse datasets, which often lead to improved accuracy in analysis tasks. As a result, FL has emerged as a promising technique that can revolutionize the way AI models are trained and deployed, all while maintaining the privacy and security of users' data. FL has distinct characteristics when compared to distributed learning. Firstly,

communication in FL is often slower and less stable. Secondly, FL involves participants with heterogeneous devices, which vary in terms of their computing capabilities. Lastly, privacy and security are emphasized more in FL. Although most studies assume that both participants and servers are trustworthy, this may not always be the case in reality.

Several studies have been conducted to offer a comprehensive understanding of the current state of FL, its potential applications, and the ongoing efforts to overcome its challenges and limitations [4–11]. These studies explore various methods and techniques, such as optimization algorithms for efficient model aggregation, privacy-preserving mechanisms, and adaptive learning strategies. While these works acknowledge the potential benefits of FL, such as collaborative ML across decentralized data sources, privacy preservation, and empowering edge devices, they predominantly focus on the theoretical aspects. However, they lack practical and experimental analysis. Therefore, further research and development are needed to incorporate more practical implementations and experimental evaluations to validate the theoretical findings and provide real-world insights into the effectiveness and scalability of existing FL systems [12–14].

However, implementing and deploying FL systems can be even more challenging due to a variety of factors such as high communication costs, heterogeneity in data, regulations, and tasks across different participating organizations, the autonomy and redundancy of processing nodes, and the potential for data poisoning incidents. This study aims to address these challenges by analyzing and validating the FL system architecture against these key issues. The contributions of this work can be categorized as follows:

(a)  Reporting and comprehensively describing the main challenges faced by FL systems: This involves identifying and discussing the challenges that FL systems are likely to encounter during deployment, including data-related challenges such as data heterogeneity and data privacy concerns, as well as challenges related to system architecture and design.

(b)  Integrating different tools in the FL system: This involves incorporating various AI-based tools and technologies into the FL system architecture to facilitate real-world FL scenarios using distributed hardware infrastructure. This could include implementing data aggregation techniques, incorporating secure communication protocols, and integrating privacy-preserving methods for data sharing and training.

(c)  Evaluating the FL system against reported challenges: This objective involves conducting dedicated experiments for different tasks and data types to evaluate the FL system's performance and accuracy against the challenges identified in objective (a). This will involve assessing the system's ability to handle data heterogeneity, maintain data privacy, and handle data poisoning incidents.

(d)  Providing an overall assessment of the developed FL system: The final objective of this work is to provide an overall assessment of the FL system architecture developed through this study. This includes analyzing the system's scalability, efficiency, and robustness against the challenges identified in objectives (a) to (c).

Overall, this study seeks to contribute to the overall understanding of the challenges faced by FL systems and provide solutions to improve their efficiency, scalability, and security. The outcomes of this study could have significant implications for the development and deployment of FL systems in various domains, including healthcare, finance, robotics and education.

The remainder of the paper is organized as follows: A comprehensive study is presented in Sections 1.1–1.3. The adopted FL framework is detailed in Section 2. Experimental results are discussed in Section 3. Research findings and limitations are highlighted in Section 4, while conclusions are drawn in Section 5.

### 1.1. Communication Efficiency and System-Specific Challenges

In developing federated networking methodologies, the communication overhead must be considered as an important constraint. Although a comprehensive review of communication-efficient distributed learning methods is beyond the scope of this article,

we can identify several general directions for addressing this issue. These directions can be roughly divided into two categories: (a) local updating methods and (b) compression schemes. Reducing the communication overhead is essential to make FL flexible against the explosive growth of datasets. To achieve this goal, reducing the number of communication rounds and improving the model upload speed have been effective efforts to further minimize the update time. Recently, several methods have been proposed to improve communication efficiency [15–18]. The communication between the server and the clients is intended to be as small as possible in order to reduce the upload time. This is achieved by allowing, in distributed settings, a variable number of local updates to be applied in parallel on each machine during each round of communication. This makes the amount of computation versus communication much more flexible. In practice, these methods drastically improve performance and have been shown to achieve significant speedups over traditional distributed approaches. For federation environments, optimizations allowing flexible local updates and low client participation are common.

In particular, the research on FL has focused on increasing communication efficiency and accelerating model updates. McMahan et al.'s [19] pioneering work introduced the concept of averaging local stochastic gradient descent updates to increase the calculated quantity of each client between communication rounds. Nishio and Yonetani's [16] FedCs framework aimed to integrate as many available clients as possible in each training round by using maximum mean discrepancy. Yurochkin et al.'s [15] Bayesian nonparametric FL framework could aggregate local models without extra parameters, achieving satisfactory accuracy with only one communication round. To accelerate model updates, structured and sketched update strategies were proposed to reduce communication pressure [19]. Jiang and Ying's [20] adaptive method for local training adjusted the local training epochs based on the server's decision, reducing local training time when the loss is small. Liu et al. [21] utilized momentum gradient descent to consider previous gradient information in each local training epoch to accelerate convergence speed. However, these algorithms may not be suitable for all federal settings, and more flexible communication-efficient methods need to be explored for high efficiency demands in digital forensics.

On the other hand, model compression schemes such as sparsification, subsampling, and quantization can significantly reduce the size of messages communicated in each round, complementing local update methods that reduce the total number of communication rounds. However, these compression schemes face challenges in federated environments, such as low device participation, non-independent and identically distributed (non-IID) local data, and local updating schemes. To address these challenges, recent works have proposed strategies such as enforcing sparsity and low-rank in updating models, using structured random rotation for quantization [18], applying lossy compression and dropout to reduce server-device communication [22], and using Golomb lossless encoding [23]. Recent studies, employ online knowledge distillation approaches, also called codistillation, for communication-efficient FL. Unlike transferring model updates, codistillation focuses on transmitting the local model prediction on a public dataset that is accessible to multiple clients. This method proves beneficial in reducing communication costs, particularly when the size of the local model exceeds the size of the public data [24–26].

Federated networks often have significant system heterogeneity, with devices varying in terms of hardware, network connectivity, and battery power. This variability can lead to unbalanced training times and introduce issues such as stragglers, which are more common in federated networks than in centralized systems. To address these issues, methods have been proposed that focus on (a) resource allocation for heterogeneous devices and (b) fault tolerance for devices that are prone to being offline. Previous works have focused on allocating resources properly to heterogeneous devices in FL. This includes motivating high-quality devices to participate [27], exploring novel device sampling policies based on systems resources [16], studying training accuracy and convergence time with heterogeneous power constraints [28], considering the impact of resource heterogeneity on training time [29], and designing fairness metrics to impel fair resource allocation [30]. However, it

is also worth considering actively sampling a set of small but sufficiently representative devices based on the underlying statistical structure. Authors in [31] address the challenges of statistical and system heterogeneity in multi-device environments, by adopting a user-centered approach. The proposed aggregation algorithm incorporates accuracy- and efficiency-aware device selection and enables model personalization to devices.

On the other side, fault tolerance becomes crucial as some participating devices may drop out before completing the training iteration. Strategies to deal with this issue include resisting device drop out through low participation [32], ignoring failed devices [33], and designing a secure aggregation protocol [34] that is tolerant against arbitrary dropouts. The literature has also taken stragglers into account and allowed devices to spend different local update computation times, utilized a cache structure to store unreliable user updates [35], and designed fault-tolerant methods for devices prone to being offline [36].

### 1.2. Data-Specific Challenges

As previously mentioned, one of the key challenges in FL is to handle heterogeneity in terms of data. In real-world scenarios, the clients usually have significant heterogeneity among their local data distributions and dataset size since each client identically collects the local data based on its own preferences, resources, and sampling pool. The non-IID data cause inconsistencies among the local models' objectives that affect the aggregation process and lead to the global model's divergence and poor performance. More specifically, due to the different data distributions, the local training algorithms have different optimal points from the global optimal point and therefore there are local drifts between the global model and the local models. The traditional machine learning approach assumes that data are identically independent, which is not the case for non-IID data. As we keep training local models with heterogeneous local data and aggregate them to produce the global model, the divergence among the parameters of these models will accumulate and will lead to a skewed global model with poor performance [37]. Based on data distribution over the sample and feature spaces, FL approaches can be typically categorized in horizontal, vertical, and hybrid schemes. In short, horizontal federated learning uses datasets with the same feature space across all devices, Vertical federated learning uses different datasets of different feature space to jointly train a global model, while hybrid FL is a combination of the first two (in terms of feature and sample distribution).

The main categories of non-IID data include (i) label distribution skew, where the distribution of labels is different across different nodes; (ii) feature distribution skew, where the distribution of features is different across different nodes; (iii) the same label but different features, where the same label is used for different features across different nodes; (iv) same features but different label, where the same features are used for different labels across different nodes; and (v) quantity skew, where the amount of data available at local nodes is different [38].

The case 'same label but different features', knows as concept drift, is mainly related to vertical FL where the clients share overlapped sample IDs with different features. However, the experiments conducted follow a horizontal FL approach, where the clients share the same feature space but own different sample IDs. Moreover, the case "same features but different label", known as concept shift, is not applicable in the presented test case as the clients share common knowledge about a task. Based on these observations, the possible non-IID data distribution types for our tools are label distribution skew, feature distribution skew, and quantity skew.

To address this problem, researchers have proposed modifying the local training mode, adding extra procedures to the data pre-processing stage, or focusing on a global model. These approaches involve different hyperparameter choices and are aimed at improving the training of a single global model on the non-IID data.

The FedAvg [19], the most common FL algorithm, does not guarantee global models' convergence under the non-IID data settings as it does not consider the inconsistency between the local and global models. There have been some studies trying to develop effective

FL strategies for non-IID data distributions by addressing the client drift. Mohri et al. [39] emphasized the importance of fairness and improved the global model to cope with a mixture of different clients. Wang, X. et al. [40] discussed the convergence behavior of FL based on gradient-descent in non-IID data background and provided different convergence theorems for FedAvg in non-IID situations. While the FedProx [5] algorithm, inspired by FedAvg, adds a proximal term to the local objective to force the local updates to be closer to the global model. FedProx was proposed as a modification to the FedAvg method to ensure convergence in practice. The algorithm has been evaluated on four datasets (https://leaf.cmu.edu/, accessed on 15 June 2023), where the data distribution among the clients imposes statistical heterogeneity. MNIST and FEMNIST datasets were used for the image classification task, the Sentiment140 for text sentiment analysis and the dataset of "The Complete Works of William Shakespeare" for next-character prediction. Moreover, FedNova [41] improves the FedAvg in the aggregation phase by normalizing and scaling the local updates of each client according to their number of local epochs to ensure that all the local updates influence the global model equally and prevent biases. The method was evaluated on a non-IID partitioned CIFAR-10 dataset (https://www.cs.toronto.edu/~kriz/cifar.html, accessed on 15 June 2023) for image classification. SCAFFOLD algorithm [42] applies a variance reduction technique to estimate the update direction of the global model and the update direction of each local model, calculate the drift of local training and correct the local updates by adding this drift to the training process. Huang, Shea et al. [43] introduced clustering thought with FL, while hierarchical heterogeneous horizontal frameworks [44] were used to overcome data heterogeneity. Data augmentation and personalization were also explored to make the data across clients more similar. Lastly, the FedOpt strategy [45] allows the usage of adaptive optimizers, including ADAGRAD, and ADAM [46] to improve the convergence of the global model.

In a recent study [47], researchers introduced a novel approach, named as MOON, where they propose a local model constrastive loss comparing representations of global and local models from successive FL rounds. This technique aims to improve the training of individual parties by conducting contrastive learning in the model-level, specifically in the feature representation space, pushing the current local representation closer to the global representation and further away from the previous local one. Similarly, authors in [48] proposed a distillation-based regularization method, named FedAlign, that promotes local learning generality while maintaining excellent resource efficiency.

In our experiments, we focused on investigating how the statistical data distribution heterogeneity affects the convergence and performance of the global aggregated model. For each tool, we built distributed local datasets with different types and levels of heterogeneity to simulate realistic FL scenarios.

### 1.3. Privacy-Related Challenges

FL systems are designed to improve data privacy since the gradient information is shared between the FL participants, while the transmission of raw data is not required. However, research studies reveal that FL does not guarantee adequate privacy and security to the system. Research has shown that even a small fraction of gradients can expose sensitive information about local data in FL, and sharing model updates during training can also pose a risk to privacy [49,50]. For instance, there is evidence about FL attacks that are able to retrieve speaker information from the transferred gradients when training FL systems for automatic speech recognition applications [51]. To mitigate these threats and ensure the privacy and security of FL, it is important to investigate potential attacks on the FL network and develop defense mechanisms. Some relevant studies provide useful strategies for protecting FL against such attacks. During the training phase of FL networks, attacks that occur are known as poisoning attacks. These attacks can impact either the dataset or the local model, with the goal of modifying the behavior of the FL network in an undesirable way and distorting the global ML model's accuracy and performance. As it is already stated, attackers can exploit the communication protocol amongst different

participants to perform malicious actions, which can target either the data of each client (data poisoning attack) or the shared model parameters (model poisoning attack). Data poisoning attacks compromise the integrity of training data, while model poisoning attacks target partial or full model replacement during training. In an FL system, attacks can be performed by either the central server or the participating clients of the FL system. Figure 1 shows that data poisoning is performed at local data collection, while model poisoning is sourced at the local model training process.

In the case of data poisoning attacks, the attacker aims to degrade the performance of a set of target nodes by injecting corrupted/poisoned data into nodes. Label flipping is a data poisoning attack in which malicious nodes change the labels of samples either arbitrarily or with a specific pattern. In the first case, a different label is randomly assigned to a sample in such a way that the global performance of the FL model is reduced. In the second case, the malicious node assigns specific labels to a set of records with a clear purpose in mind. Due to the vast number of participants involved in the FL system, it is not guaranteed who is an honest as well as a credible entity and who is a malicious actor.

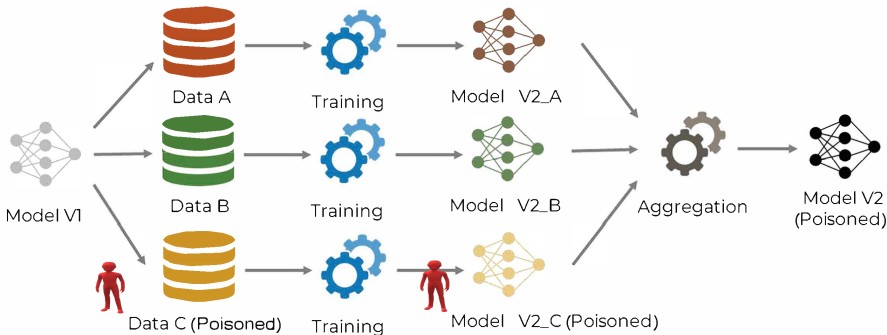

**Figure 1.** Data vs. model poisoning attacks on federated learning (FL) system. In the first case the data of the third node has been compromised in such a way as to affect the training process. In the second case, the attacker is trying to interfere with the main training process. In the present work, we focused on the first scenario.

The impact of data poisoning attacks in image classification tasks is investigated in [52] using the benchmark dataset MNIST and CIFAR-10. Specifically, this work studies the impact of data poisoning attacks on FL models regarding various percentages of malicious participants, random and targeted label flipping, and the time of the attack. A higher percentage of malicious nodes results in higher degradation of the model performance. Moreover, a targeted data poisoning attack is detected more difficult. Finally, the time that an attack is performed is a crucial aspect, while a model that is trained with malicious nodes up to a point can converge if enough time is given. The paper in [53] presents two variants of data poisoning attack, namely model degradation and targeted label attacks. Both of those attacks are based on synthetic images generated by generative adversarial networks (GANs). Through experiments, it is observed that the GAN-based attacks manage to fool common federated defenses. Specifically, the model degradation attack provokes around 25% accuracy degradation, while the targeted label attack results in label misclassification of 56%. The authors also introduce a mechanism to mitigate these attacks, which is based on clean label training on server side. A distributed backdoor attack as a data poisoning attack on FL systems is proposed in [54]. A local trigger is chosen by each adversary instead of a common global one. During inference, attackers exploit the local triggers to form global ones. This work compares this kind of attack to a centralised approach, and they conclude that it is more persistent than the centralised scenarios.

The design of a robust defence mechanism against data poisoning attacks in FL systems is a challenging task since most of them are attack-specific that have been designed for a specific type of attack and they do not work well for other types [55–57]. The case of non-IID data among FL clients introduces other challenges in the procedure of developing

an efficient defence mechanism against data poisoning attacks. Specifically, the work in [58] presents that the non-IID data increases the difficulty of an accurate defence against data poisoning attacks. When the data are not identically distributed among FL participants, each client has its own data distribution, therefore bringing its unique contribution to the common FL model, which is misleading for most defence mechanisms. Regarding the model poisoning attacks, participants' private information can be extracted from the sharable weights throughout the training process. Therefore, sensitive information can be revealed either to third-party or to the central server.

Many mechanisms have been developed to enhance the privacy of FL, such as secure multiparty computation (SMC) or differential privacy (DP). The aforementioned techniques provide privacy for the cost of decreased ML model performance. A challenge that should be faced during the development of a secure FL system is understanding and balancing the trade-off between the privacy-preserving level and the achieved ML performance. The authors in [59] evaluates the performance of the proposed FL system under various settings of differential privacy as a privacy preserving technique and configurations of the FL nodes, investigating the trade-off between those components and achieved model performance. Specifically, they demonstrate how the model performance is affected at different level of DP, when the number of participants increases as well as in the case of imbalanced client data. Beyond providing an adequate level of privacy, it is also essential to implement computationally cheap FL methods, communication efficient as well as tolerant dropped devices without compromising accuracy.

The various privacy approaches of an FL system can be grouped into two categories, which are global privacy and the local privacy. In the former, the central server is a trusted party and the model updates generated at each round are considered as private to all untrusted participants except the central server. In the latter, all the participants may be malicious, and, therefore, the updates are also private to the server. A very common approach to prevent leakage of private client data from the shared parameters is the utilization of the secure aggregation mechanism of FL. In the last few years, various FL designs have been introduced that use secure aggregation protocols under various setups. Bonawitz in [60] introduces a secure aggregation mechanism for FL, which can tolerate client dropouts. This method uses Shamir's Secret Sharing [61] and symmetric encryption to prevent the server from accessing individual model updates. The limitation of this approach lies in the fact that it requires at least four communication rounds between each client and the aggregator in each iteration. Bonawitz's protocol is utilized by the works VerifyNet [62] and VeriFL [63], which add an extra verification level on top of [60]. The additional verifiability guarantees the correctness of the aggregation. However, those methods require a trusted party to create private keys for all clients. Trying to reduce the overhead of [60], the algorithm in [64,65] present secure aggregation mechanisms with polylogarithmic communication and computation complexity. The main differences in [60] are that those methods replace the star-like topology of the FL network with random subgroups of clients as well as the secret sharing is only performed for a set of clients and not for all of them. Both methods [64,65] demand three rounds of communication interaction between the server and the clients. Another approach that reduces the communication and computation overhead compared to [60] is the Turbo-Aggregate [66]. Specifically, this method utilizes a circular communication topology. The FastSecAgg [67] method uses Fast Fourier transform multi-secret sharing for secure aggregation. FastSecAgg is robust against adversaries which adaptively corrupt clients during the execution of FL procedures.

Several survey works have been published in the last past years with the aim to review and summarize the latest papers related on adversarial attacks and threats on FL system as well as the possible defense mechanisms. The authors in [68] present an extensive review of the various threats that can be applied to a FL system as well as their corresponding countermeasures. Specifically, they provide several taxonomies of adversarial attacks and their defense methods, depicting a general picture of the vulnerabilities of FL and how to overcome them. The threats that can introduce vulnerabilities to trustworthy FL system,

across different stages of the development procedure are introduced in [69]. This work analyses the attacks that can be performed by a malicious participant in FL during data processing, model training, deployment and inference. Additionally, the authors of this paper aim to assist on the selection of the most appropriate defense mechanism by discussing specific technical solution to realize the various aspects of trustworthy FL. The work [70] provides a review to the concept of FL, threat models, and two major attacks, namely poisoning attacks and inference attacks, by highlighting the intuitions, key techniques, and fundamental assumptions of the attacks. Some years after, a more comprehensive survey on privacy and robustness in FL is conducted and presented in [71]. The authors of this work review the various threat models, privacy attacks, and poisoning attacks as well as their corresponding defenses. Finally, the paper [72] demonstrates a comprehensive study regarding the security and privacy aspects that need to be considered in a FL setup. The results obtained from this research indicate that communication bottlenecks, poisoning, and backdoor attacks are the most specific security threats, while the inference-based attacks are the most crucial to the FL privacy.

## 2. Materials and Methods

### 2.1. FL Topologies and Design Principles

FL involves distributing the training process across a network of nodes, and the arrangement of this network can significantly impact the model training. The various topologies used in FL differ in factors such as the aggregation algorithm and distribution of the model, the number of communication links needed for training, and the associated costs of setting up the training system. It is possible to implement FL with or without a central server for orchestration. Three common network topologies used in FL are the *star* topology, *ring* topology, and *hybrid* topology, as documented in Ganapathy's work [73]. In a *star-like topology* for FL, each node communicates with a central server and only shares small updates to the model with the server. This allows for asynchronous learning, where nodes can train independently without waiting for other nodes. Adding new nodes is straightforward, but the speed of training can vary due to differences in device configurations and the number of training samples. Each node requires two communication links for each round of communication: one to upload local updates and another to download the global model from the server. In the *ring-like topology* for FL, devices are connected in a circular data path where each node is connected to two others. The training process starts with the first node, and a global model is initialized across all clients. Each node updates the model weights after a fixed number of local iterations and transfers it to the next adjacent node. This cyclical weight transfer is repeated until convergence, and the output of the last node is the final model that needs to be redistributed among all clients. This approach follows synchronous learning, as nodes depend on their peer nodes, and computational capabilities should be comparable across all nodes. As this topology does not require a central server, the cost of infrastructure development is lower, and fewer communication links are needed. The *hybrid topology* combines the star-like and ring-like topologies by involving both nodes and a server. Nodes are grouped together and arranged sequentially, and the first node in each group begins training asynchronously using their local data. Each node sends its model updates to the next adjacent node in the group after training for a pre-defined number of local epochs. This process is repeated for all nodes in each group, following the ring-like topology. The hybrid topology uses a combination of synchronous and asynchronous learning, and the computational power of the server can be compromised compared to the star-like topology. If the number of nodes is equal to the number of groups, it is equivalent to FL in star-like topology.

### 2.2. FL Platform Processes and Workflow

Before starting FL, the data needs to be annotated and split into training and validation sets. A tool is needed to establish communication protocols and orchestrate the various tasks necessary for federated training. This tool retrieves code files, executes the training

process, and stores the results in an experiment tracking component. The specific tool also acts as a model registry, storing the resulting model version in a central repository. The best performing model is chosen and fused with its previous version using a state-of-the-art weight aggregation method, defined during the configuration of the orchestration component. Lastly, the fused model is then served back to the FL platform, ready for deployment. The aforementioned process is illustrated in Figure 2.

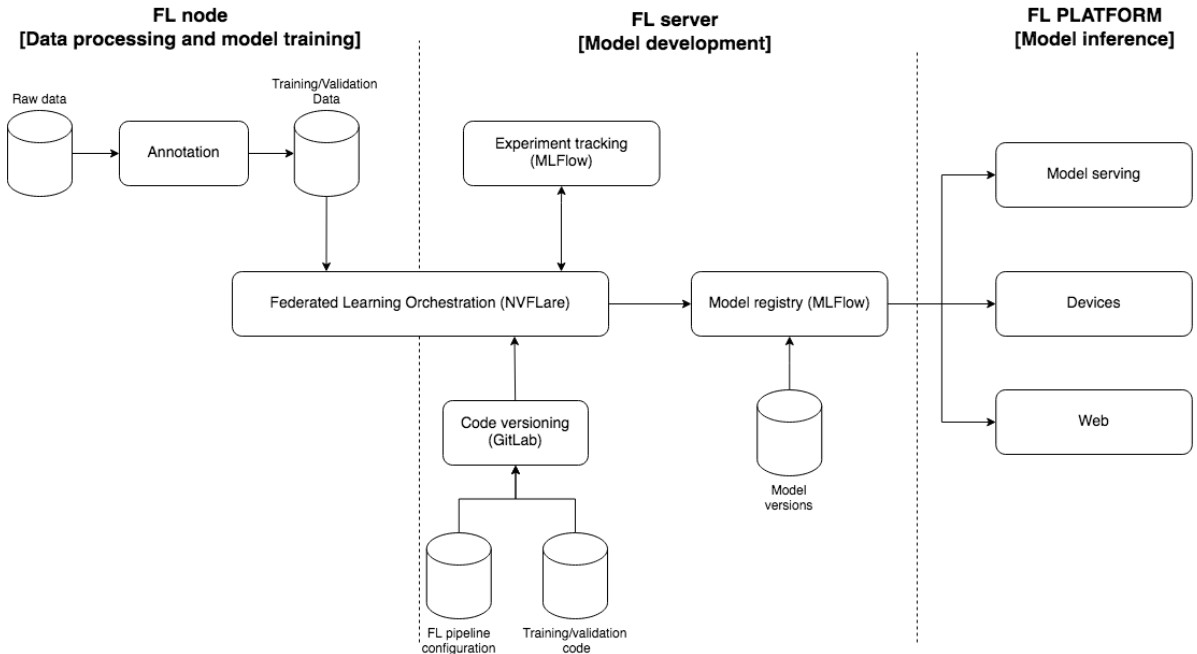

**Figure 2.** Building -Block View of FL Platform.

Figure 3 shows the workflow for FL, which involves configuring and deploying the central orchestration tool to establish connections between nodes and ensuring information security. Annotated data are placed in a specific directory, and the data scientist organizes code for tasks such as data preparation and model training. Once new training data are in the data owner's directory, FL orchestration tool is triggered, and the model is sent to the data owner party. After training, metadata and artifacts are ingested into the ML lifecycle management tool. The FL server merges the locally trained models of each node utilizing a weighted aggregation method. The aggregated global model and validation script are sent back to the data owner party for evaluation, and results are assessed by the data scientist. If necessary, the loop starts over with a new training. If the model performs well, it is stored in the model registry for deployment in inference.

*2.3. FL Framework*

The FL orchestration process refers to the coordination activities performed while applying learning technologies. The orchestrator is the actor in FL who is responsible for accepting and forwarding device connections through a secure communication protocol while taking into consideration certain participation criteria such as the optimal number of the participating devices, fast internet connection speed, etc. It is responsible for handling failures in the FL process, including connected devices crashing to ensure the training will continue to make progress. Most of the orchestration strategies assume full devices partici-pation; that is, all devices participate in every training round. In practice, the participation of all devices does not guarantee the convergence of the FL training process. An efficient orchestrator must take into consideration insights regarding each device's heterogeneous properties, considering data heterogeneity and bias, computational resources, and Internet connection in order to implement the most efficient device selection strategy.

## Federated Learning Server workflow

**Figure 3.** Workflow view of FL.

The NVFLARE framework (https://nvidia.github.io/NVFlare/, accessed on 15 June 2023) has been selected for the development of the FL platform as it supports both deep learning and traditional ML algorithms, as well as horizontal and vertical FL. It includes built-in FL aggregation algorithms (e.g., FedAvg, FedProx, FedOpt, Scaffold, Ditto) and supports multiple training and validation workflows (i.e., global model evaluation, cross-site validation), data analytics, and ML lifecycle management. NVFLARE also offers privacy preservation with differential privacy and homomorphic encryption as well as security enforcement through federated authorization and privacy policy. It is easily customizable and extensible and can be deployed on cloud and on premise. Additionally, NVFLARE includes a simulator for rapid development and prototyping, a dashboard user interface (UI) for simplified project management and deployment, and built-in support for system resiliency and fault tolerance.

ML Lifecycle Management: The ML lifecycle involves six processes, including Experimentation and Prototyping, which is essential for achieving the best model. Model management is necessary to handle different model-variants and experiments, track ML metadata, and govern model deployment. MLflow (https://mlflow.org/, accessed on 15 June 2023) is an open-source tool for ML lifecycle management that consists of four components, including MLflow Tracking and MLflow registry. MLflow enables remote monitoring of training logs, tracking of hyperparameters, evaluation of models against quality measures and fairness indicators, central storage and retrieval of models, continuous evaluation of deployed models, and traceability, debugging, and reproducibility of potential issues. For FL, MLflow tracking and MLflow registry were deployed, providing a UI that facilitates the interaction with the MLflow server.

### 2.4. Infrustructure and Implementation Details

Regarding the hardware infrastructure, for the FL experiments, we used five physical nodes located at different locations across the EU (Spain, Greece, Portugal, and Cyprus). Following the star-like topology (see Figure 4), every local node connects to the central server and establishes a one-to-one communication. In our case, a central server has been selected among the five nodes that act as the aggregator server for all the FL experiments and the other devices are the clients that hold the local datasets. Except for the physical nodes, we created an extra virtual client (*dummy* node) that contains a global dataset and is used only for evaluation purposes. Overall, we have one aggregator server, five nodes for both training and evaluation, and one virtual node only for validation. Secure communication between clients and the server is established through a virtual private network (VPN) connection to ensure that sensitive data (parameters) are safely transmitted and to prevent unauthorized access. All the devices that participate in the FL process are almost identical and have the same resources. The aggregator server has five Nvidia RTX 3090 GPUs with 24$GB$ VRAM, two nodes (Spain) have four Nvidia RTX 3090 GPUs with 24$GB$ VRAM, one node (Greece) has four Nvidia RTX $A$5000 GPUs with 24$GB$ VRAM, and the last two nodes (Cyprus and Portugal) are equipped with four Nvidia RTX 3090 GPUs and 24$GB$ VRAM.

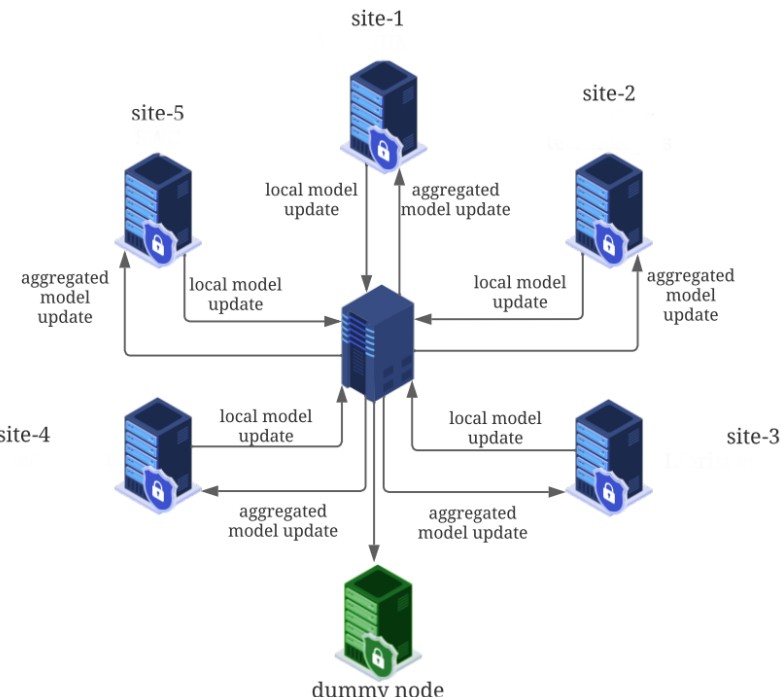

**Figure 4.** The adopted star-like topology utilizes a centralized server to create and share the global model, while five peripheral nodes participate asynchronously in the training process.

*2.5. Workflow Adaptations*

Unlike previous studies that conduct mostly simplified proof-of-concept (PoC) experiments in federated settings, usually involving virtual nodes, in this work we report the results and the analysis of large-scale FL experiments for various AI tools, using the specific hardware infrastructure described under realistic and demanding scenarios. Since we conducted the experiments under realistic conditions in physical nodes, we follow a more sophisticated FL approach than starting the process, training the models, and evaluating the performance. Firstly, the network configuration file is created to provide information related to participants, including domain names, port numbers, or ID addresses that will be used for the connection with the server. For each different tool's experiments, we use a separate set of ports to avoid conflicts during the models training in parallel. Each developer creates for each of his/her tools a Dockerfile that contains the required information and packages for the communication establishment with the aggregator and the deployment of the tools to clients' sites. The Dockerfile is sent to each client device through the VPN connection to be built. Once all the clients have the required information, the FL process starts and follows the FL server's and clients' configurations.

The NVFLARE framework offers built-in methods and algorithms but also gives the capability to build custom methods dedicated to specific needs. In our case, our tools cover a large range of different tasks and data types, including *face re-identification (Face ReID)*, *video super-resolution (VSR)*, *named entity recognition (NERC)*, and *audio speech recognition (ASR)* and is necessary to develop some custom methods to handle the local training and evaluation and the global aggregation process.

**FL Server (Aggregator) Stack:** As we mentioned before, we created a virtual (*dummy*) node only for evaluation purposes. The node holds a global dataset that is used to evaluate the performance and generalization of both the locally trained and the global aggregated models. Since the '*dummy*' node does not participate in the training process, we built a custom aggregation method that can identify and exclude the weights of the *Dummy* node from the aggregation process. Moreover, the built-in JSON generator function, which is used to report the results of the FL process in a JSON format, was modified to enable the usage of MLFlow for the tracking of the training and evaluation phases by logging parameters, code versions, metrics, and output files. Last, besides the evaluation of the local models on their local test sets and the global test set, we also conducted the cross-site evaluation, when all the clients have finished training. During the cross-site model evaluation, every client validates other clients' models and the global model on their local datasets.

**FL Client (Data Owner) Stack:** The client configuration files contain mainly all the parameters that are required for the local training process, including model and data loader arguments, number of local epochs, etc. These parameters are different for each tool, and the developer is responsible to choose the more suitable ones and create the client configuration files. The only extra arguments that have been added are a few parameters related to the MLFlow framework for result reporting.

## 3. Results

*3.1. FL Strategies*

Section 1 presents a detailed description of the FL data-specific challenges related to local data distribution and dataset size heterogeneity. To tackle data heterogeneity in the FL process, multiple FL algorithms and strategies have been developed, including more sophisticated aggregation methods and strategies for more efficient local training. In our experiments, we evaluate the performance and the convergence of the global aggregated model by utilizing the three more well-known and widely used FL strategies, FedAvg, FedOpt, and FedProx, representing both the classic baselines and current state of the art. Each of these methods focuses on handling data heterogeneities considering label distribution skew, feature distribution skew, and quantity skew, which are highly related to

datasets of the reported AI tools, exploring the effectiveness of these methods on different types of data sources such as images, audio, and text.

### 3.2. FL Security Mechanisms

To enhance privacy preservation in the FL system, a DP method is applied as a security mechanism during the experiments. DP is one of the most well-known and widely used approaches for privacy preservation, which randomizes part of the system's behavior to provide privacy. Additionally, with DP the users can quantify the level of privacy of the system by selecting the appropriate parameters that achieve the best trade-off between the FL model performance and privacy level. Specifically, at tests the SVT (sparse vector technique) privacy protocol is utilized since it is characterized as a fundamental method to perform DP. The SVT privacy is applied as a filter in the FL set up of NVFlare, while the definition of it as well as its necessary parameters are performed by the configuration file of the client. The chosen parameters of the SVT privacy method are listed in Table 1 for each experiment.

**Table 1.** Parameters of sparse vector technique (SVT) privacy for each experiment.

| Experiment ID | Fraction | Epsilon | Noise_var |
| --- | --- | --- | --- |
| Face-ReID | 0.6 | 0.1 | 1 |
| VSR | 0.6 | 0.001 | 0.1, 1 |
| NERC | 0.6 | 0.001, 0.1 | 0.1, 0.5, 1 |
| ASR | 0.6 | 0.001, 0.1 | 0.1, 0.5, 1 |

### 3.3. Data Management

A typical lifecycle of ML models entails their training on example data. In the typical workflow, a model is trained on large amounts of local data and tested for performance on a smaller disjoint dataset. The training itself comprises several rounds of model runs (i.e., inference) and adjustments in order to converge to an acceptable performance. On the other hand, FL is a distributed ML paradigm where different sets of data, typically disjoint among them, are used at multiple self-sustained training locations, i.e., training nodes. A schematic representation is drafted in Figure 5. Different versions of local models are fine-tuned at each federated party, while the final global model is obtained after their aggregation.

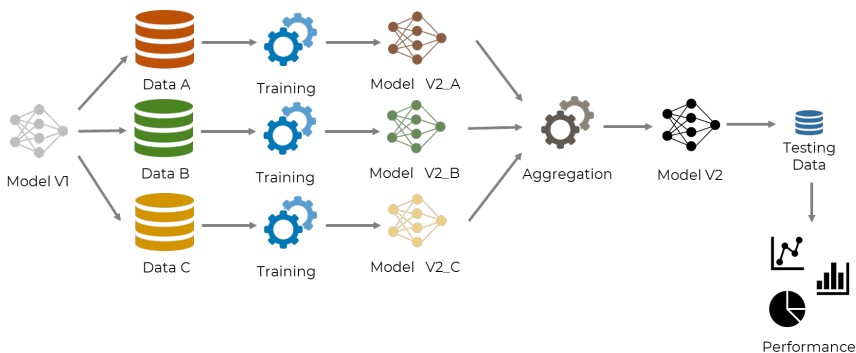

**Figure 5.** Schematic workflow of a basic FL principle.

In the FL setting, however, the aggregation of multiple models attenuates the differences among the local models and thus their adaptation to the local data. Multiple rounds of local training are necessary to reach convergence of the global model. The adaptation of a model can thus be tracked at each round. The ML training workflow uses a distinct set of data to train or assess the evolution of the training process. We distinguish between:

- **Training set:** A set of examples used for learning, i.e., fitting the best parameters of the ML model (classifier, detector, etc.). This is the set of positive and negative examples available at each federated party.
- **Validation set:** A set of examples, the disjoint from the training set, used to assess the model's actual performance at each epoch of the local training process. It is used by the researchers to examine the training process for possible irregularities and adjustment of the training optimization parameters. Similarly, it can be used by researchers to tune any hyper-parameters of an ML model, such as the number of hidden units in a neural network or any other specific internal settings.
- **Test set:** A set of examples used only to assess the performance of a fully specified classifier. During the FL setup, the different sets introduced beforehand must be defined at each federated party, as is illustrated in Figure 6.

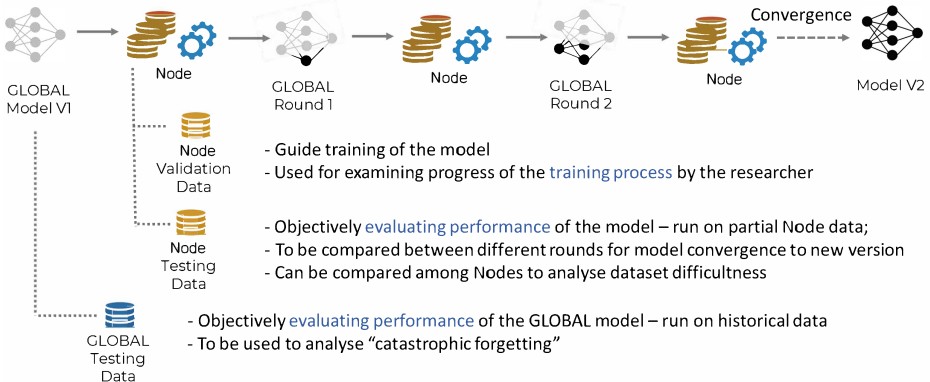

**Figure 6.** The use of different datasets during the FL training.

In the terms of the FL testing of this manuscript, we pursue the testing on local nodes data as well as on the GLOBAL dataset in each round. While the GLOBAL testing set evaluates a possible drift in the performance of the initial task, the local datasets give us a good estimation of how the models are adapting to new data. Moreover, two options for testing on local data are available: (a) testing before the local training, i.e., evaluating the external knowledge on the local conditions and (b) testing after the local training, i.e., evaluating the adaptability of the global model to the local conditions. With respect to the testing of the global model, these two options refer to (a) testing the global model after aggregation and (b) testing the global model before aggregation. We use the MLFlow environment to track the training performance. Overall, this environment can be used simultaneously by the researchers as well as possible stakeholders to obtain a common view on the training process (see Figure 7), i.e., monitoring the training loss/accuracy at every round and each federated node by the researchers, or monitoring the overall performance of the global model by the end-users.

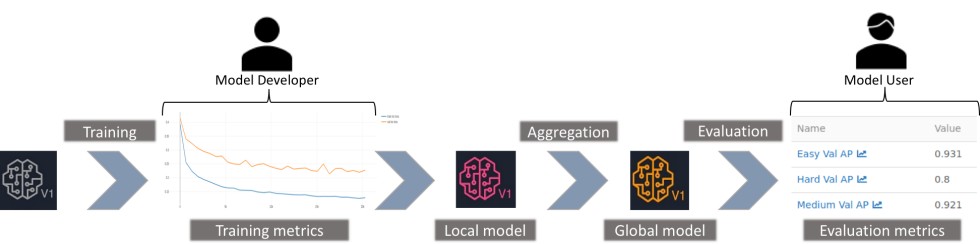

**Figure 7.** The use of MLFlow environment for tracking the training process.

### 3.4. Experimental Evaluation

#### 3.4.1. Face Re-ID

**Datasets:** In the federated setting, experiments on the face re-identification tool involve using different datasets for both training and testing the model at each federated node.

These datasets are carefully selected to cater to various scenarios and final use cases. The training dataset is the *Ms1m-Retinanet*, a public dataset published in [74]. The dataset is derived from the Ms1m dataset, which comprises images of celebrities. To obtain a cleaned version, the images were cropped to a size of $112 \times 112$ using Retinanet and five facial landmarks. A pre-trained model based on ArcFace [75] and ethnicity-specific annotators was then used for semi-automatic refinement. Ultimately, our training dataset consisted of around 50,000 identities, which were equally distributed among the federated nodes, with 10,000 identities per node. The experiments were performed using five nodes.

The goal of our experiment was to evaluate the adaptability of the face re-identification model to face occlusions within a federated framework, under multiple heterogeneity settings. To create an occluded version of the dataset, the *EyesOcclusionGenerator* library was used to add black rectangles over the eyes of the identities in the original dataset. The final training dataset comprised these two versions of the dataset, with and without eye occlusions. During training, the image of the identity was randomly selected from one of the datasets, with a certain probability.

To introduce heterogeneity in the experiments, one of the two datasets (Ms1m-Retinanet with and without occlusions) was exclusively used for training in some of the nodes. This was achieved by adjusting the probability of selecting the dataset for the next batch during training. The specific configuration per traning node is given in Table 2. The training was conducted from scratch, with the initial random model being adapted to the previously unseen examples, and executed over 8 rounds with 10 epochs.

**Table 2.** Training data distributions among clients for different levels of heterogeneity.

| Heterogeneity | Site-1 | Site-2 | Site-3 | Site-4 | Site-5 |
|---|---|---|---|---|---|
| *Low* | 50% [1] | 50% | 50% | 50% | 50% |
| *Medium* | 50% | 50% | 50% | 100% | 0% |
| *High* | 50% | 100% | 100% | 0% | 0% |

[1] Represents the percentage of occlusion in the training set.

During the testing phase, and similarly to the test setting of the ArcFace model [75], we employed multiple datasets, such as $LFW$, $CFP$, and $Age-DB-30$ for checking the convergence on slightly different domain. The test datasets consisted of both the original as well as eye-occluded versions. For the sake of simplicity, $LFW$ denoted as $D^1$, $CFP-FP$ denoted as $D^2$, $CFP-FF$ denoted as $D^3$, and $Age-DB-30$ denoted as $D^4$ (with their occluded versions denoted as $D^1_{occl}$, $D^2_{occl}$, $D^3_{occl}$, and $D^4_{occl}$, respectively). The occluded versions of the testing datasets were produced using the same library used for the training dataset. These testing datasets were distributed across the individual training nodes, with each node managing a distinct testing dataset as shown in Table 3.

**Table 3.** Testing data distributions among clients.

| Site-1 | Site-2 | Site-3 | Site-4 | Site-5 | Dummy |
|---|---|---|---|---|---|
| $D^1$ | $D^2$ | $D^2$ | $D^1$ | $D^3$ | $D^4$ |
| $D^1_{occl}$ | $D^2_{occl}$ | $D^2_{occl}$ | $D^1_{occl}$ | $D^3_{occl}$ | $D^4_{occl}$ |

**Quantitative:** We test the face re-identification model performance using the validation protocols of well-known public face-recognition datasets (i.e., verification performance of a pair of images). The results of the experiments for the face re-identification tool are shown in the following Table 4.

**Table 4.** Verification performance evaluation of the centrally trained and the FL aggregated model on the global and the local datasets, where *low*, *medium*, and *high* denote the level of heterogeneity.

| Dataset | Centralised | $FedAvg_{low}$ | $FedAvg_{medium}$ | $FedAvg_{high}$ | $FedAvg_{DP}$ [1] |
|---|---|---|---|---|---|
| $D^4$ | 0.978 | 0.852 | 0.85 | **0.858** | N/A |
| $D^4_{occl}$ | 0.834 | **0.708** | 0.686 | 0.683 | N/A |
| $D^3$ | 0.998 | **0.966** | **0.966** | 0.964 | N/A |
| $D^3_{occl}$ | 0.936 | 0.812 | 0.818 | **0.826** | N/A |
| $D^2$ | 0.974 | 0.88 | 0.88 | **0.885** | 0.806 |
| $D^2_{occl}$ | 0.781 | 0.703 | 0.701 | **0.704** | 0.642 |
| $D^1$ | 0.997 | 0.968 | 0.967 | **0.978** | 0.946 |
| $D^1_{occl}$ | 0.969 | 0.881 | **0.891** | 0.887 | 0.839 |

[1] Equally split dataset and svt_privacy.

It should be noted that the last configuration used for the distributed training encountered issues with the connectivity of some of the nodes, resulting in a lack of results for some testing datasets. The provided Figure 8a–c offer a comparison between the local training models obtained from each node and the global model for each heterogeneity experiment. The accuracy of each model was evaluated on the local test dataset, with identical results for nodes 1 and 4 as well as 3 and 5. The graphs suggest that there is a discernible variance in the accuracy of models trained on nodes that solely utilize occluded data versus those that do not, as per the various heterogeneity settings outlined in Table 2. This difference is more evident in Figure 9a,b, which depict non-occluded and occluded images, respectively.

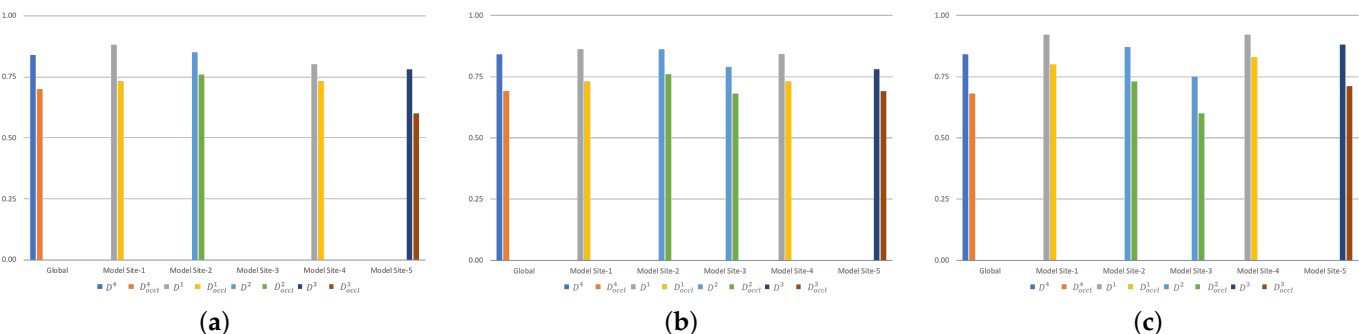

(**a**)  (**b**)  (**c**)

**Figure 8.** (**a**) Low-, (**b**) medium-, and (**c**) high-heterogeneity results of global and local models at each node (ordered as dummy and nodes 1 to 5). Note that the results for node 3 are missing within the low heterogeneity setting (**a**).

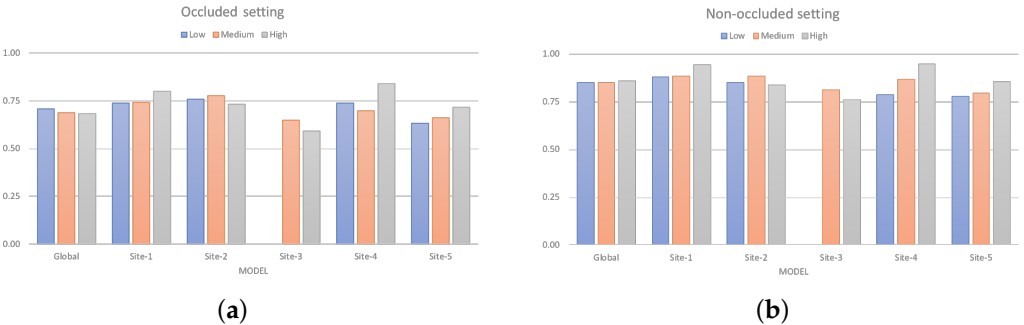

(**a**)  (**b**)

**Figure 9.** Testing results comparing different heterogeneity settings, for (**a**) occluded and (**b**) non-occluded images. The results are grouped and ordered from dummy to nodes 1 to 5.

The graphs provided above enable us to compare the accuracy of the local models from each node within the local dataset across the different heterogeneity settings outlined in

Table 5. Site 1's model was trained with 50% occluded data in each heterogeneity experiment, resulting in similar results. However, minor differences suggest the influence of other nodes on the training of the aggregated model utilized in each round. Site 2, in contrast, trains solely with occluded data in the high heterogeneity experiment, resulting in a slight performance decay. Site 3 shows a similar pattern to Site 2, although the low-heterogeneity setting was incomplete due to technical connectivity problems. Site 4's model is trained solely with occluded data in the medium-heterogeneity experiment and with only non-occluded data in the high-heterogeneity experiment. This results in better performance on the non-occluded version of the dataset, but worse on the occluded version in the medium-heterogeneity experiment. However, in the high-heterogeneity case, the model performs better in both the non-occluded and occluded versions of the testing dataset. Finally, Site 5 trains exclusively with non-occluded data in the medium- and high-heterogeneity experiments. The model performs better on both the non-occluded and occluded versions of the testing dataset compared to training with 50% occluded data.

**Qualitative:** The experimental results indicate that the FL framework produces inferior results compared to local training. The variance in accuracy in some cases may be due to longer training times without the FL framework. Interestingly, the results for different data heterogeneities were similar, suggesting that heterogeneity had minimal impact on training. The $D^1$ testing dataset achieved the highest accuracy for both the occluded and non-occluded versions. Regarding the security preserving mechanism experiment, due to node connectivity issues, the results cannot be accurately compared to previous experiments since the number of nodes was reduced. However, it was observed that the model converged, which is not the case with other tools such as NERC under similar circumstances. It is important to note that the results for this experiment were only obtained for some of the testing datasets.

**Table 5.** Training data distributions for different levels of heterogeneity.

| Heterogeneity | Site-1 | Site-2 | Site-3 | Site-4 | Site-5 | *Dummy* |
|:---:|:---:|:---:|:---:|:---:|:---:|:---:|
| *Low* | 5470 | 5967 | 4560 | 4177 | 4122 | N/A |
| *High* | 11,470 | 4072 | 824 | 1344 | 6490 | N/A |
| *Validation* [1] | 760 | 624 | 471 | 462 | 347 | 1934 |

[1] Validation data distribution is the same for both heterogeneity scenarios.

### 3.4.2. Video Super-Resolution

**Datasets:** For the VSR task, the dataset used was the Vimeo90k [76]. This dataset is of high quality and contains a large variety of scenes and actions, with video clips of a fixed resolution of $448 \times 256$. The upscaling factor used for the experiments was set to 4. The data heterogeneity type that was considered for the VSR task was quantity skew. This type of heterogeneity aims to simulate unbalanced data distribution by allocating a different number of video clips among the FL clients. To explore different levels of size heterogeneity, various splitting scenarios were designed.

Table 5 presents the data distributions among the five clients based on different levels of size heterogeneity. The clients are named as Site-1, Site-2, and so on, with each corresponding to a specific server. The *dummy* node is included only for evaluation purposes and does not contain any training data. In the low-level heterogeneity case, all clients have a similar number of low-, high-resolution (LR-HR) video clip pairs for training. However, in the high heterogeneity case, the number of samples for each client can be significantly different, resulting in an imbalanced distribution of data.

Two types of evaluation datasets were created for the VSR task: local and global evaluation datasets. Each FL client has its own local validation/test dataset, which is distinct from the local training and validation datasets of other clients. Additionally, a global validation dataset was created, located on the '*dummy*' node, which contains video clips with varied content and actions and a more balanced distribution for evaluating the performance and generalization of both locally trained models and the globally aggregated model. The table

displays the number of samples per local dataset and the size of the global dataset, which remain constant across different levels of heterogeneity.

**Quantitative:** The VSR task was evaluated using the FedAvg FL method, and the results are presented in Table 6. The evaluation was based on the peak signal-to-noise ratio (PSNR) metric (dB) on the global dataset for both the centrally trained model and the globally aggregated model for different levels of heterogeneity. The Vimeo90k dataset was used for the experiments, with an upscaling factor of 4 and the data heterogeneity type being quantity skew. The locally trained models were evaluated on their respective local validation/test datasets, while a global validation dataset was used to evaluate the performance and generalization of both the locally trained models and the globally aggregated one. The centrally trained model had the highest performance and served as the baseline. The FL global model, obtained by aggregating the local models' weights, achieved similar performance to the central training, with the performance gap being wider in the high heterogeneity case. These results indicate that the FL approach can successfully handle low to medium level of data distribution heterogeneity utilizing a simple aggregation scheme (FedAvg), but for higher level of heterogeneity where the gap between the different local optimal points is larger, more sophisticated learning strategies are needed.

**Table 6.** Performance comparison between the centrally trained model w/o FL (baseline) and the globally aggregated model on the global dataset for different FL strategies and high level of heterogeneity.

| Heterogeneity | Centralised | $FedAvg_{global}$ | $FedProx_{global}$ | $FedOpt_{global}$ |
|---|---|---|---|---|
| *Low* | 36.15 | 35.87 | - | - |
| *High* | 36.15 | 35.60 | 35.63 | 35.71 |

To address the issue of high heterogeneity in the size of local datasets, more advanced FL algorithms and strategies were employed in the experiments. Specifically, the *FedOpt* and *FedProx* methods were used. These two learning strategies have been extensively used to tackle heterogeneity in federated networks, requiring minor modifications in the aggregation scheme, enabling an easy integration into the existing framework. FedProx adds a regularization term to the local objective minimization algorithm to reduce the shift between the local and global objectives, while FedOpt enables the use of adaptive optimizers on the server's side to improve model convergence. In our experiments, we want to investigate if these two long-established methods that follow different approaches can handle a realistic FL scenario in practice. Table 6 provides a comparison of the evaluation results for the different FL strategies on the global VSR dataset. As shown, the FedOpt strategy can more efficiently handle the quantity skewness than the FedProx, which achieves a very small improvement. This result can be justified by assuming that the regularization term of the FedProx method is more suitable for objective shifts resulting from label or feature distribution skew. Furthermore, we conducted an experiment to explore the impact of a privacy-preserving mechanism, specifically the SVT method, on the performance of the global model (see Table 7). In this experiment, a low-level heterogeneity approach was used to investigate the effects of the privacy mechanism only on the global model's performance. The results, presented in the table below, show that the SVT method with a noise_var of 0.1 reduces the model's performance by almost $3dB$. This indicates that it is crucial to strike the best balance between privacy and performance. A high level of privacy, as achieved with a noise_var of 0.1, may prevent any information leakage but can result in a significant decrease in performance.

**Table 7.** Effect of SVT mechanism in global model's performance.

| Heterogeneity | $FedAvg_{global}$ | $FedProx_{global+SVT}$ [1] | $FedOpt_{global+SVT}$ [2] |
|---|---|---|---|
| *Low* | 35.87 | 34.45 | 32.50 |

[1] noise_var = 1, [2] noise_var = 0.1.

The upcoming experiments focus on the impact of the FL processing on the performance of the local models. Table 8 displays the performance of the local models and the globally aggregated model on the local validation datasets throughout the FL training, comprising 12 rounds. The table showcases the difference in performance at each round.

**Table 8.** Performance comparison of global and local models on the local datasets. Experiment: High level of heterogeneity, FedOpt strategy, w/o privacy-preserving mechanism.

| Client | Model | 1 | 2 | 3 | 4 | 5 | 6 | 7 | ... | 11 | 12 |
|---|---|---|---|---|---|---|---|---|---|---|---|
| Site-3 | *Local* | 34.23 | 34.75 | 35.03 | 35.32 | 35.44 | 35.50 | 35.56 | | 35.72 | 35.74 |
| | *Global* | 34.64 | 27.90 | 32.65 | 35.03 | 34.97 | 36.02 | 36.28 | | 36.68 | 36.70 |

Through the analysis of Table 8, we can gain a deeper understanding of how the FL process operates. Initially, for at least the first seven rounds of FL, the local models outperform the global aggregated model on their respective local datasets. This is because the local models prioritize minimizing the loss for their specific datasets, whereas the global model has a more general objective. However, as the FL training process continues, the global model can handle the different objectives of the local models through the aggregation algorithm, eventually achieving better performance than the local models on the local datasets. The benefits of FL training are more pronounced for clients with limited samples and limited performance capabilities, such as Site-3, where the aggregated model outperforms the local one by 0.12 dB.

**Qualitative:** This sub-section summarizes the key insights and conclusions derived from the experiments on FL VSR. Firstly, the FL global model can achieve performance comparable to the ideal centrally trained model, demonstrating the potential of using distributed training with data available in multiple devices. Secondly, the heterogeneity in data distribution, caused by variations in the number of samples per client, negatively impacts the performance of the global model. However, the use of FL strategies developed to handle heterogeneous data can limit this impact. Thirdly, certain privacy-preserving mechanisms, such as SVT, can lead to a significant reduction in the FL model's performance, highlighting the need to find an optimal balance between performance and privacy. Lastly, FL can be beneficial for clients lacking sufficient data to build high-performance models.

### 3.4.3. Named Entity Recognition and Classification

**Datasets:** The NERC (named entity recognition and classification) uses data from CoNLL datasets, which are widely recognized in the research community. These datasets contain sentences taken from news articles that are manually labeled with different types of entities, such as PER (person), LOC (location), ORG (organization), and MISC (miscellaneous). To introduce heterogeneity among the FL clients, two strategies have been followed: (a) language heterogeneity and (b) size heterogeneity. In terms of language heterogeneity, examples from English, Spanish, and Dutch have been combined and unevenly distributed across the subsets used by the clients, resulting in some clients having more examples from one language than others. As for size heterogeneity, different numbers of documents have been scattered across the FL clients, causing some clients to have significantly more training examples than others, simulating a possible scenario in which some of the participating clients have little training data in comparison to the others. The following table illustrates the dataset distributions achieved by implementing the aforementioned heterogeneity approaches. The FL clients are denoted as Site-1, Site-2, and so on. Additionally, a *dummy* node is included as a client with no training data, solely intended for evaluation purposes

and containing data that does not belong to any client. The Tables 9 and 10 show how for language heterogeneity strategy the distribution of training instances per language varies from client to client, while for size heterogeneity strategy, it is the distribution of the total number of training instances (while keeping a balance language-wise) which varies among clients.

**Table 9.** NERC datasets distribution for the language unbalance heterogeneity strategy.

| Sets | Language | Site-1 | Site-2 | Site-3 | Site-4 | Site-5 | *Dummy* |
|------|----------|--------|--------|--------|--------|--------|---------|
| *train* | es | 4995 | 3442 | 0 | 1396 | 0 | N/A |
| | en | 0 | 2294 | 8424 | 2793 | 0 | N/A |
| | nl | 0 | 1147 | 0 | 4415 | 9483 | N/A |
| | total | 4995 | 6884 | 8424 | 8381 | 9483 | N/A |
| *validation* | es | 455 | 583 | 0 | 395 | 0 | 1355 |
| | en | 0 | 291 | 1381 | 395 | 0 | 1355 |
| | nl | 0 | 291 | 0 | 791 | 1558 | 1355 |
| | total | 455 | 1167 | 1381 | 1583 | 1558 | 4065 |

**Table 10.** NERC datasets distribution for the size unbalance heterogeneity strategy.

| Sets | Language | Site-1 | Site-2 | Site-3 | Site-4 | Site-5 | *Dummy* |
|------|----------|--------|--------|--------|--------|--------|---------|
| train | es | 5089 | 2544 | 2544 | 1272 | 1272 | N/A |
| | en | 5089 | 2544 | 2544 | 1272 | 1272 | N/A |
| | nl | 5089 | 2544 | 2544 | 1272 | 1272 | N/A |
| | total | 15,269 | 7633 | 7633 | 3816 | 3816 | N/A |
| validation | es | 507 | 507 | 507 | 507 | 507 | 846 |
| | en | 507 | 507 | 507 | 507 | 507 | 846 |
| | nl | 507 | 507 | 507 | 507 | 507 | 846 |
| | total | 1523 | 1523 | 1523 | 1523 | 1523 | 2540 |

**Quantitative:** In the context of NERC, an experiment was conducted involving five clients and one validation *dummy* node. The validation node contained more balanced data, while the five clients had different levels of data imbalance to simulate heterogeneity. The goal of the experiment was to evaluate NERC performance in the presence of data heterogeneity. Tables 11 and 12 show the main results obtained when comparing both size and language-based unbalance heterogeneity levels in one of the nodes, and the dummy node, which is balanced. The evaluation is measuring the F-score over the detected named entities, which is the harmonic mean of the precision (the fraction of relevant entities among the retrieved entities) and recall (the fraction of relevant instances that were retrieved).

**Table 11.** NERC FL experiments with the language unbalance heterogeneity strategy. The performance is measured using the F-score.

| | FL round (only for FL) | 1 | 5 | 10 | 15 | 20 | 25 |
|--|--|--|--|--|--|--|--|
| | Training steps | 600 | 3000 | 6000 | 9000 | 12,000 | 15,000 |
| Test data | Evaluated model | | | | | | |
| *Dummy* data | Local Training (baseline) | - | 0.857 | 0.886 | 0.883 | 0.889 | 0.895 |
| Site-5 data | Local Training (baseline) | - | 0.863 | 0.881 | 0.886 | 0.898 | 0.895 |
| *Dummy* data | Aggr. FL model | 0.757 | 0.845 | 0.856 | 0.857 | 0.862 | 0.856 |
| Site-5 data | Aggr. FL model | 0.701 | 0.783 | 0.804 | 0.814 | 0.824 | 0.809 |

**Table 12.** NERC FL experiments with the size unbalance heterogeneity strategy. The performance is measured using the F-score.

| Test data | FL round (only for FL) Training steps Evaluated model | 1 600 | 5 3000 | 10 6000 | 15 9000 | 20 12,000 | 25 15,000 |
|---|---|---|---|---|---|---|---|
| *Dummy* data | Local Training (baseline) | - | 0.854 | 0.885 | 0.881 | 0.887 | 0.892 |
| Site-5 data | Local Training (baseline) | - | 0.874 | 0.895 | 0.898 | 0.908 | 0.910 |
| *Dummy* data | Aggr. FL model | 0.751 | 0.833 | 0.842 | 0.841 | 0.840 | 0.844 |
| Site-5 data | Aggr. FL model | 0.766 | 0.812 | 0.834 | 0.842 | 0.855 | 0.841 |

The experimental results were positive, with high scores achieved by the different nodes despite the presence of data heterogeneity. In addition, privacy-preserving mechanisms were tested using SVT (see Table 13), and the obtained results for various SVT parameterizations are presented in a table below. The experiments focused on heterogeneity based on differences in data size, and some cells in the table are empty due to errors encountered during certain parameter configurations.

**Table 13.** NERC FL experiments with SVT privacy (language unbalance data).

| | Test data | FL round (only for FL) Training steps Evaluated model | 1 600 | 5 3000 | 10 6000 | 15 9000 | 20 12,000 | 25 15,000 |
|---|---|---|---|---|---|---|---|---|
| $SVT_{fraction:0.6,epsilon:0.001,noise\_var:1}$ | *Dummy* data | Aggr. FL model | 0.021 | 0.021 | 0.021 | - | - | - |
| | Site-5 data | Aggr. FL model | 0.722 | 0.717 | 0.723 | - | - | - |
| $SVT_{fraction:0.6,epsilon:0.001,noise\_var:0.5}$ | *Dummy* data | Aggr. FL model | 0.029 | 0.029 | 0.029 | 0.029 | 0.029 | 0.029 |
| | Site-5 data | Aggr. FL model | 0.706 | 0.740 | 0.710 | 0.728 | 0.686 | 0.679 |
| $SVT_{fraction:0.6,epsilon:0.001,noise\_var:0.1}$ | *Dummy* data | Aggr. FL model | 0.019 | 0.019 | 0.019 | 0.019 | 0.019 | 0.019 |
| | Site-5 data | Aggr. FL model | 0.730 | 0.742 | 0.728 | 0.726 | 0.726 | 0.718 |
| $SVT_{fraction:0.6,epsilon:0.1,noise\_var:1.0}$ | *Dummy* data | Aggr. FL model | 0.016 | - | - | - | - | - |
| | Site-5 data | Aggr. FL model | 0.723 | - | - | - | - | - |

Looking at Table 14, it appears that the federated model becomes excessively noisy and ineffective for the *Dummy* node, although the reason for this outcome is unclear. Additionally, a data-poisoning experiment was conducted in which one of the nodes was provided with 'poisoned' data. Specifically, the training data for this node was manipulated to switch all the labels for individuals (PER) and locations (LOC). The validation data for this node remained unaltered. This experiment was conducted using the size-unbalance heterogeneity strategy and the performance using measured using the F-score.

**Table 14.** Data-poisoning experiment, with one of the clients having 'poisoned' training data.

| | FL round (only for FL) Training steps | 1 600 | 5 3000 | 10 6000 | 15 9000 | 20 12,000 | 25 15,000 |
|---|---|---|---|---|---|---|---|
| *Dummy* data | Aggr. FL model | 0.756 | 0.817 | 0.788 | 0.831 | 0.837 | 0.811 |
| Site-1 data (poisoned) | Aggr. FL model [1] | 0.288 | 0.408 | 0.408 | 0.396 | 0.414 | 0.411 |
| Site-1 data (poisoned) | Aggr. FL model [2] | 0.76 | 0.844 | 0.796 | 0.838 | 0.849 | 0.831 |
| Site-2 data | Aggr. FL model [1] | 0.783 | 0.843 | 0.847 | 0.848 | 0.852 | 0.852 |
| Site-5 data | Aggr. FL model [1] | 0.721 | 0.833 | 0.827 | 0.822 | 0.83 | 0.843 |

[1] plus 1 round of local training, [2] without further local training.

**Qualitative:** Based on the experimental results of the NERC tool using FL, several conclusions can be drawn. Firstly, data heterogeneity does not pose a significant problem, indicating that FL is viable in these situations. Both scenarios of heterogeneity devised are realistic and demonstrate the potential of FL to combine different data sources. Secondly, the privacy-preserving mechanism used in the experimentation (SVT) does not seem to be suitable for the NERC model, leading to a drop in performance to that of a randomly

initialized model. Thirdly, the federated training approach is robust against data poisoning, as it enables learning across datasets from different stakeholders and results in a more coherent and robust model. Overall, the findings suggest that FL can be a valuable approach for NERC models, allowing for joint learning across disparate data sources and robustness against data poisoning, while careful consideration should be given to the choice of privacy-preserving mechanism.

### 3.4.4. Audio Speech Recognition

**Datasets:** The ASR data consider a different dataset from the literature for each client. The considered corpora include the TEDLIUM [77] (node-1), debating technologies [78] (node-2), Librispeech-other (node-3), Librispeech-clean (node-4) [79], and the Spoken Wikipedia Corpus (https://nats.gitlab.io/swc/, accessed on 15 June 2023) (node-5). The data configuration is designed to evaluate the impact of heterogeneity levels on data distribution, which is more realistic for the application. To ensure equal data amounts in each node, the number of samples from each corpus is adjusted, and each node is given a subset of 1400 speech recordings for local training. For experiments with low heterogeneity levels, some samples from TEDLIUMv2 and the Spoken Wikipedia Corpus are replaced by samples from debating technologies and librispeech-other datasets, respectively. The AI used for the pipeline is based on a Wav2Vec2.0 model [80], which is a self-supervised end-to-end architecture based on convolutional and transformer layers. The training hyperparameters were the same for the five nodes, and include a batch size of 2, a learning rate of $5 \times 10^{-5}$ warmed up in the first 10% of the training time, and a gradient accumulation of 16 steps. The local training is performed for 5 epochs. The central server is configured to run for 10 rounds of federated training.

**Quantitative:** The experiments conducted on ASR include (1) low heterogeneity, (2) high heterogeneity, (3) high heterogeneity with an SVT privacy-preserving strategy, and (4) high heterogeneity with a percentile privacy (PP)-preserving strategy. The low-heterogeneity experiment aims to evaluate the capabilities of the system to recognize speech under relative controlled acoustic conditions and with noise-controlled levels. On the contrary, the high-heterogeneity experiment introduces datasets with high level of noise and in non-controlled acoustic environments. For the SVT and PP, we evaluate several hyperparameters (see Table 1) to test the impact of the privacy preserving approaches. The main results of these experiments are presented in Table 15, where the performance is evaluated based on the word error rate (WER) after 10 rounds of federated training in each node.

**Table 15.** Summary of FL results for the different configurations of data heterogeneity and privacy-preserving approaches.

| Experiment | Description | WER | | | | | |
| --- | --- | --- | --- | --- | --- | --- | --- |
| | | Site-1 | Site-2 | Site-3 | Site-4 | Site-5 | Dummy |
| 1 | Low heterogeneity | 17.7 | 15.3 | 8.1 | 2.7 | 25.8 | 3.9 |
| 2 | High heterogeneity | 13.5 | 12.4 | 7.9 | 2.8 | 25.7 | 3.8 |
| 3 | SVT [1] | 19.7 | 15.3 | 7.8 | 2.9 | 24.0 | 3.6 |
| 4 | PP [2] | 19.7 | 15.3 | 7.8 | 3.0 | 24.0 | 3.6 |

[1] fraction = 0.6, eps = [0.001, 0.001, 0.001, 0.1], noise_var = [1, 0.5, 0.1, 1], [2] perc = [10, 40, 70], gamma = 0.01.

Table 16 provides further details on the experimental results for each of the four ASR experiments. The tables present information on the behavior of federated training over 10 rounds as well as a comparison between the Word Error Rate (WER) before and after aggregation on local test sets.

**Table 16.** Federated training in different heterogeneity and privacy settings for *dummy* node and Site-1. Rows 1 and 2 correspond to low and high heterogeneity, respectively, and rows 3 and 4 correspond to the high-heterogeneity setting for SVT and PP settings, respectively.

| | Node | Aggr. | 1 | 2 | 3 | 4 | 5 | 6 | 7 | 8 | 9 | 10 |
|---|---|---|---|---|---|---|---|---|---|---|---|---|
| 1 | Site-1 | Before | 19.7 | 18.3 | 17.9 | 18.6 | 19.0 | 19.0 | 19.5 | 19.5 | 19.4 | 19.5 |
| | | After | 19.9 | 18.2 | 18.2 | 17.9 | 17.9 | 17.8 | 18.2 | 17.9 | 17.8 | 17.8 |
| | Dummy | After | 3.6 | 3.7 | 3.7 | 3.8 | 3.8 | 3.9 | 4.0 | 4.0 | 3.9 | 3.9 |
| 2 | Site-1 | Before | 13.6 | 13.6 | 13.5 | 13.5 | 13.4 | 13.4 | 13.4 | 13.3 | 13.3 | 13.2 |
| | | After | 13.5 | 13.6 | 13.5 | 13.5 | 13.5 | 13.5 | 13.3 | 13.2 | 13.2 | 13.2 |
| | Dummy | After | 3.9 | 3.8 | 3.8 | 3.7 | 3.7 | 3.7 | 3.6 | 3.6 | 3.6 | 3.6 |
| 3 | Site-1 | Before | 19.8 | 19.7 | 19.7 | 19.7 | 19.7 | 19.7 | 19.7 | 19.7 | 19.7 | 19.7 |
| | | After | 19.6 | 19.7 | 19.6 | 19.7 | 19.6 | 19.7 | 19.6 | 19.7 | 19.6 | 19.7 |
| | Dummy | After | 3.6 | 3.6 | 3.6 | 3.6 | 3.6 | 3.6 | 3.6 | 3.6 | 3.6 | 3.6 |
| 4 | Site-1 | Before | 19.8 | 19.7 | 19.7 | 19.7 | 19.7 | 19.7 | 19.7 | 19.7 | 19.7 | 19.7 |
| | | After | 19.7 | 19.7 | 19.7 | 19.7 | 19.7 | 19.7 | 19.7 | 19.7 | 19.7 | 19.7 |
| | Dummy | After | 3.6 | 3.6 | 3.6 | 3.6 | 3.6 | 3.6 | 3.6 | 3.6 | 3.6 | 3.6 |

**Qualitative:** The experiments for speech recognition have provided several insights. Firstly, data heterogeneity did not have a significant impact on the performance of the speech recognizer, as observed in the *dummy* node's results. This suggests that FL can be used to obtain a joint and potentially richer model by combining sources of data that cannot be otherwise combined. Secondly, the use of the SVT and PP privacy-preserving approaches did not facilitate any improvement in the model during the federated setting. After the first round, the results for different nodes did not change (see Table 16, presenting only Site-1 states for simplicity), indicating that these approaches do not help in learning anything useful for the local model. Note that in the same, high-heterogeneity setting, without privacy preserving schemes, the WER were changing throughout the FL process, finally reaching lower values.

## 4. Discussion

Summarizing the findings, it could be highlighted that the system adaptation pipeline is capable of enabling the integration of different (in terms of modality, domain, and task) AI-based tools into the FL system and the FL training under realistic conditions. However, the results showed that the performance of the FL model may be affected by certain challenges, such as data heterogeneity and the use of specific privacy-preserving mechanisms. This is highlighted in the low-heterogeneity experiment of Table 16, where the WER decreases sequentially in Site-1 after 10 aggregation rounds, unlike the other experiments. This can be explained by the IID conditions in Experiment 1 (i.e., the data exhibits IID characteristics), where the central model benefits from the other nodes' contributions, leading to a reduction in WER specifically in Site-1. However, in experiments with higher heterogeneity, the lack of dataset representativeness prevents the central model from achieving similar improvements in Site-1's WER. Although there have been numerous studies on FL recently, there is a lack of research exploring the effectiveness of presented aggregated methods on different types of data sources such as images, audio, and text when used with deep learning models. Our experiments demonstrate that these existing studies offer minimal or no advantage over the traditional DL approaches. To mitigate these challenges, appropriate FL strategies need to be selected, such as aggregation algorithms that handle heterogeneous data and privacy mechanisms that strike a balance between performance and privacy. Moreover, no substantial problems were identified for the integration of the tools into the FL framework, except in some specific cases where the type of ML model hindered any development.

In terms of privacy, the integrated security mechanisms seem to provide a satisfactory level of protection against privacy attacks, although it is important to find the best trade-off between performance and privacy. In particular, even though in most cases the chosen security strategies had no particular impact on the training of the models, specific privacy-preserving mechanisms (e.g., SVT) could dramatically decrease the FL model's performance. In terms of fairness, the chosen aggregation mechanisms seem to mitigate the potential biases of the ML model, ensuring that the FL system does not yield systematic advantages to certain privileged nodes. For example, the results in Table 4 for the occluded datasets ($D_{occl}^1$, $D_{occl}^2$, $D_{occl}^3$, and $D_{occl}^4$), the centrally trained model's accuracy drops compared to the global datasets. However, the FL aggregated models, particularly in the case of high hetergoneity, still exhibit respectable performance, indicating their ability to handle occluded data under realistic conditions.

Regarding robustness, the results showed that the FL system appears to be resistant to data poisoning attacks. More specifically, FL appears to be resistant to label-based data poisoning attacks (intentional or not) and, according to the experiments, even nodes with poisoned data can benefit from the resulting federated model. In particular, results in Table 14 suggest that despite the poisoned data in Site-1 (the client has intentionally injected poisoned training data), the aggregated FL model manages to achieve comparable performance to the models trained on clean data from other sites (Site-2 and Site-5). This indicates the resilience of the FL system against label-based data poisoning attacks. However, it is important to test the system's robustness under different conditions, including naturally occurring conditions and those set up by malicious actors.

Finally, as a final observation, we could say that the global FL model can achieve similar performance to the ideal model trained centrally, taking into account the characteristics of each node, and each type of data and selecting the appropriate training tools (i.e., aggregation algorithms and privacy mechanisms). All the above indicate that FL platform has the potential to progress from proof of concept to a trustworthy application, as long as it is tested and assessed against appropriate performance indicators in a precise context.

## 5. Conclusions

This study addresses the challenges that may arise when using analytic AI-based tools in a distributed and realistic environment, specifically under a modular FL platform. We have presented an insightful analysis of the federated learning key characteristics and adapted the system architecture, workflows, pipelines, and training schemes to the FL hardware requirements at each federation node under realistic conditions. We selected a representative set of AI-based tools that cover all possible validation cases, taking into account the size, the type, and the format of the data required for their training. In addition, we shaped and distributed the datasets to each edge device of the federation so that the scenarios we validate will reflect the real-world cases as closely as possible. Finally, the developed FL system was evaluated concerning the aforementioned challenges by conducting specific experiments for different tasks and data types, providing an overall assessment, following both a qualitative and quantitative approach. The findings highlight the capability of the system adaptation pipeline to integrate diverse AI-based tools into the FL system and train under realistic conditions while facing technical limitations and significant challenges due to data heterogeneity, privacy, and security concerns. The study emphasizes the importance of selecting appropriate FL strategies, such as aggregation algorithms and privacy-preserving mechanisms, to address challenges related to performance, privacy, fairness, and robustness, thereby enabling FL to become a trustworthy and effective application.

**Author Contributions:** Conceptualization, P.D., C.Z.P., and A.P.; methodology, A.P., P.L., and A.D.; software, J.C.V.-C., A.G.-P., S.C.S., K.Z., and S.B.; validation, A.P., P.L., S.B., and K.Z.; formal analysis, A.P., A.D., and P.L.; investigation, A.P.; resources, J.C.V.-C., A.G.-P., S.C.S., K.Z., and P.L.; data curation, K.Z. and A.P.; writing—original draft preparation, A.P., J.C.V.-C., A.G.-P., S.C.S., K.Z., and S.B.; writing—review and editing, A.P., A.D., P.L., P.D., and C.Z.P.; visualization, K.Z. and S.B.;

supervision, P.D. and C.Z.P.; project administration, P.D. All authors have read and agreed to the published version of the manuscript.

**Funding:** This research was funded by European Commission under contract H2020-883341 GRACE.

**Data Availability Statement:** No new data were created in this study. Data sharing is not applicable to this article.

**Conflicts of Interest:** The authors declare no conflict of interest.

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
