# Peer review of "Fostering Trustworthiness of Federated Learning Ecosystem through Realistic Scenarios"

_information, doi:10.3390/info14060342_

Round 1

Reviewer 1 Report

Very comprehensive article with detailed experiments in realistic environment. Moreover, authors considered multiple trustworthiness scenarios including data management and privacy. The datasets are also comprehensively adopted across raw, image and video data. 

Discussion section could be more fulfilled which denotes the interesting finding through the experiment such as Row 1 in Table 16, the figures dropped down from 19.9 to 17.8 after aggregation while others are relatively stable. 

Authors also need to state the reason to select specific method/s have been selected during experiments stage, why and why not. 

Some more literature reviews may consider to add, especially some similar works which also works on comprehensive experimental review on FL. 

The reference list seems a bit outdated, only  very small amount was from recent two years. The research on FL and ML has been exposed in recent years. 

Minor gramma check May required

Reviewer 2 Report

Great job

Still after reading the analysis, it is not clear to me:

1. what is Sized-unbalance heterogeneity strategy and using F-score? This might be problematic for the reader as well.

2. the conclusion does not clearly support the discussion and can be improved

Round 2

Reviewer 1 Report

My concerns have been comprehensively addressed.